# NAS-Bench-360: Benchmarking Diverse Tasks for Neural Architecture Search

**Renbo Tu**
Carnegie Mellon University
renbo@cmu.edu

**Mikhail Khodak**
Carnegie Mellon University
khodak@cmu.edu

**Nicholas Roberts**
Carnegie Mellon University
ncrobert@andrew.cmu.edu

**Maria-Florina Balcan**
Carnegie Mellon University
ninamf@cs.cmu.edu

**Ameet Talwalkar**
Carnegie Mellon University and Determined AI
talwalkar@cmu.edu

## Abstract

Most existing neural architecture search (NAS) benchmarks and algorithms prioritize performance on well-studied tasks, focusing on computer vision datasets such as CIFAR and ImageNet. However, the applicability of NAS approaches in other areas is not adequately understood. In this paper, we present **NAS-Bench-360**, a benchmark suite for evaluating state-of-the-art NAS methods on less-explored datasets. To do this, we organize a diverse array of tasks, from classification of simple deformations of natural images to predicting protein folding and partial differential equation (PDE) solving. Our evaluation pipeline compares architecture search spaces of different flavors, and reveals varying performance on different tasks, providing baselines for further use. All data and reproducible evaluation code are open-source and publicly available. The results of our evaluation show that current state-of-the-art NAS methods often struggle to compete with simple baselines and human-designed architectures on the majority of tasks in our benchmark. At the same time, they can be quite effective on a few individual, understudied tasks. This demonstrates the importance of evaluation on diverse tasks to better understand the usefulness of different approaches to architecture search and automation.

## 1 Introduction

Neural architecture search (NAS) aims to automate the design of deep neural networks, ensuring performance on par with hand-crafted architectures while reducing human labor devoted to tedious architecture tuning [8]. With the growing number of application areas of ML, and thus of use-cases for automating it, NAS has experienced an intense amount of study, with significant progress in search space design [3, 20, 33], search efficiency [22], and search algorithms [16, 28, 29]. While the use of NAS techniques may be especially impactful in under-explored or under-resourced domains where less expert help is available, the field has largely been dominated by methods designed for and evaluated on benchmarks in computer vision [6, 20, 30]. There have been a few recent efforts to diversify these benchmarks to settings such as vision-based transfer learning [7] and speech and language processing [13, 21]; however, evaluating NAS methods on such well-studied tasks using traditional, domain-specific search spaces does not give a good indication of their utility on more far-afield applications, which have often necessitated the design of custom neural operations [4, 19].

We aim to rectify this issue by introducing a suite of diverse benchmark tasks drawn from various data domains that we collectively call **NAS-Bench-360**. This benchmark consists of an organized

Submitted to the 35th Conference on Neural Information Processing Systems (NeurIPS 2021) Track on Datasets and Benchmarks. Do not distribute.

setup of five suitable datasets that can both (a) be evaluated in a unified way using existing NAS approaches and (b) come from a variety of different application areas, including numerical analysis, organic chemistry, and medical imaging. We also include standard image classification evaluations as a point of comparison, as many new methods continue to be designed for such tasks.

Following our construction of this benchmark, we evaluate three different NAS approaches, each characterized by an architecture (search space, search algorithm) pair, and compare the results to expert-driven domain-specific architecture design. As a baseline comparator, the first approach uses a singleton architecture, the Wide ResNet (WRN) [31], as the search space, paired with a hyperparameter tuning algorithm to adjust the training procedure for each task. The other two search spaces are well-studied in modern NAS: DARTS [20] and DenseNAS [9], and we pair them with their respective best-performing search methods. We find that the modern NAS approaches struggle to beat even the simple WRN comparator on the majority of tasks in the benchmark. On two of the tasks—classifying electromyography signals and solving partial differential equations—NAS methods do significantly worse. NAS lags even further behind when we include domain-specific expert-designed architectures, where it lags far behind on even CIFAR-100 when disallowing extra augmentation or pre-training on ImageNet [25]. On the other hand, DARTS cells perform relatively well on two tasks that *a priori* seem more challenging: spherical image classification and protein-distance prediction. These observations and other empirical insights demonstrate the necessity of a benchmark that provides a diverse array of data domains for evaluating NAS methods. Our evaluation results also serve as a baseline for comparison in future development of NAS.

To ensure the availability and impact of this benchmark, the associated datasets and evaluation pipelines will remain open-source and accessible at `https://rtu715.github.io/NAS-Bench-360/`. Reproducibility is assured from open-sourcing all relevant code for the end-to-end procedure, including data processing, architecture search, model retraining, and hyper-parameter tuning frameworks.

## 2   Related Work

Benchmarks have been very important to the development of NAS in recent years. This includes standard evaluation datasets and protocols, of which the most popular are the CIFAR-10 and ImageNet routines used by DARTS [20]. Another important type of benchmark has been tabular benchmarks such as NAS-Bench-101 [30], NAS-Bench-201 [6], and NAS-Bench-1Shot1 [32]; these benchmarks exhaustively evaluate all architectures in their search spaces, which is made computationally feasible by defining simple searched cells. Consequently, these benchmark cells are less expressive than the DARTS cell [20], often regarded as the most powerful search space in the cell-based regime. Notably, our benchmark is *not* a tabular benchmark, i.e. we do *not* evaluate every architecture from a fixed search space; rather, the focus is on the organization of a suite of tasks to evaluate both NAS methods and search spaces, which would necessarily be restricted if we first fixed a search space to construct a tabular benchmark from.

While NAS methods and benchmarks have generally been focused on computer vision, recent work such as AutoML-Zero [23] and XD operations [24] has started moving towards a more generically applicable set of tools for AutoML. However, even more recent benchmarks that do go beyond the most popular vision datasets have continued to focus on well-studied tasks, including vision-based transfer learning [7], speech recognition [21], and natural language processing [13]. Our aim is to go beyond such areas in order to evaluate the potential of NAS to automate the application of ML in truly under-explored domains. One analogous work to ours in the field of meta-learning is the Meta-Dataset benchmark of few-shot tasks [27], which similarly aimed to establish a wide-ranging set of evaluations for that field.

## 3   NAS-Bench-360: A Suite of Diverse and Practical Tasks

In this section, we introduce the NAS setting being targeted by our benchmark, our motivation for organizing a new set of diverse tasks as a NAS evaluation suite, and our task-selection methodology. We report evaluations of specific algorithms on this new benchmark in the next section.

## 3.1 Neural Architecture Search: Problem Formulation and Baselines

For completeness and clarity, we first formally discuss the architecture search problem itself, starting with the extended hypothesis class formulation [16]. Here the goal is to use a dataset of points $x \in \mathcal{X}$ to find parameters $\mathbf{w} \in \mathcal{W}$ and $a \in \mathcal{A}$ of a parameterized function $f_{\mathbf{w},a} : \mathcal{X} \mapsto \mathbb{R}_{\geq 0}$ that minimize the expectation $\mathbb{E}_{x \sim \mathcal{D}} f_{\mathbf{w},a}(x)$ for some test distribution $\mathcal{D}$ over $\mathcal{X}$; here $\mathcal{X}$ is the input space, $\mathcal{W}$ is the space of model weights, and $\mathcal{A}$ is the set of architectures. For generality, we do not require the training points to be drawn from $\mathcal{D}$ to allow for domain adaptation, as is the case for one of our tasks, and we do not require the loss to be supervised. Note also that the goal here does not depend on the issue of computational or memory efficiency, which we do not focus on in our evaluations; there our restriction is only that the entire pipeline can be run on an NVIDIA V100 GPU.

Notably, this formulation makes no distinction between the model weights $\mathbf{w}$ and architectures $a$, treating both as parameters of a larger model. Indeed, the goal of NAS may be seen as similar to model design, except now we include the design of an (often-discrete) *architecture space* $\mathcal{A}$ such that it is easy to find an architecture $a \in \mathcal{A}$ and model weights $\mathbf{w} \in \mathcal{W}$ whose test loss $\mathbb{E}_{\mathcal{D}} f_{\mathbf{w},a}$ is low using a search algorithm. This can be done in a one-shot manner—simultaneously optimizing $a$ and $\mathbf{w}$—or using the standard approach of first finding an architecture $a$ and then keeping it fixed while training model weights $\mathbf{w}$ for it using a pre-specified algorithm such as tuned stochastic gradient descent (SGD).

This formulation also includes non-NAS methods by allowing the architecture search space to be a singleton. When the sole architecture is a standard and common network such as WRN [31], this yields a natural baseline with an algorithm searching for training hyperparameters, not architectures. On the other hand, when $\mathcal{A}$ contains a single domain-specific architecture, such as a spherical convolutional neural network (CNN) [4], it yields the "human baseline" competitor approach without search. For our empirical investigation, we compare the performance of state-of-the-art NAS approaches against that of the two singleton baselines.

## 3.2 Motivation and Task Selection Methodology

Curating a diverse, practical set of tasks for the study of NAS is our primary motivation behind this work. We observe that past NAS benchmarks focused on the creation of larger search spaces and more sophisticated search methods for neural networks. However, the utility of these search spaces and methods are only evaluated on canonical computer vision datasets. Whether these new methods can improve upon non-NAS baselines remains an open question. This calls for the introduction of new datasets lest NAS research overfits to the biases of CIFAR-10 and ImageNet. By identifying these possible biases, future directions in NAS research can be better primed to suit the needs of practitioners, thereby incentivizing the deployment of NAS techniques on real applications.

NAS-Bench-360 comprises tasks from existing datasets their variants as summarized in Table 1. This work focuses exclusively on datasets with 2d input data including images, wave spectra, differential equations, and protein sequence features. Although in practice neural networks are employed to analyze different data modalities, the most well-studied NAS approaches only accept 2d inputs and therefore we study tasks within this scope. During the selection of tasks, breadth is our main consideration. First, we formalize the categorization of tasks into **point prediction (point)** and **dense prediction (dense)** [26], respectively referring to tasks with scalar outputs and 2d matrix outputs. In other words, point prediction tasks are classification tasks, and dense prediction tasks are element-wise prediction tasks, which is a specific form of regression. The heavy bias of previous NAS research towards point prediction tasks motivates the inclusion of dense prediction tasks in our benchmark. Second, breadth is achieved by selecting tasks from various subjects and applications of deep learning, where introducing NAS could improve upon the performance of handcrafted neural networks.

## 3.3 List of Tasks from Diverse Data Sources

In lieu of providing raw data, we perform data pre-processing locally and store the processed data on a public Amazon Web Service's S3 data bucket with download links available on our website. Our data treatment largely follows the procedure defined by the researchers who provided them. This would enhance the reproducibility of results by ensuring the uniformity of input data for different pipelines. Specific pre-processing and augmentation steps are described below.

Table 1: Information of tasks in NAS-Bench-360

| Task name | Dataset size | Type | Learning objective | New to NAS |
|-----------|--------------|------|--------------------|-----------|
| CIFAR-100 | 60K | Point | Classify natural images into 100 classes | |
| Spherical | 60K | Point | Classify spherically projected images into 100 classes | ✓ |
| NinaPro | 3956 | Point | Classify sEMG signals into 18 classes corresponding to hand gestures | ✓ |
| Darcy Flow | 1100 | Dense | Predict the final state of a fluid from its initial conditions | ✓ |
| PSICOV | 3606 | Dense | Predict pairwise distances between residuals from 2d protein sequence features | ✓ |

### 3.3.1 CIFAR-100: Standard Image Classification

As a starting point of comparison to existing benchmarks, we include the **CIFAR-100** task [14], which contains RGB images from natural settings to be classified into 100 fine-grained categories. CIFAR-100 is preferred over CIFAR-10 because it is more challenging and suffers less from over-fitting in previous research.

**Data pre-processing:** while the 10,000 testing images are kept aside only for evaluating architectures, the 50,000 training images are randomly partitioned into 40,000 for architecture search and 10,000 for validation. On all of the 50,000 training images, we apply standard CIFAR augmentations including random crops and horizontal flipping, and finally normalize them using a pre-calculated mean and standard deviation of this set. On the 10,000 testing images, we only apply normalization with the same constants.

### 3.3.2 Spherical: Classifying Spherically Projected CIFAR-100 Images

To test NAS methods applied to natural-image-like data, we consider the task of classifying spherical projections of the CIFAR-100 images, which we call the **Spherical** task. In addition to scientific interest, spherical image data is also present in a variety of applications, such as omnidirectional vision in robotics and weather modeling in meteorology, as sensors usually produce distorted image signals in real-life settings. To create a spherical variant of CIFAR, we project the planar signals of the CIFAR images to the northern hemisphere and add a random rotation to produce spherical signals for each individual channel following the procedure specified in [4]. The resulting images are 60*60 pixels with RGB channels.

**Data pre-processing:** with the same split ratios CIFAR-100, the generated spherical image data is directly used for training and evaluation without data augmentation and pre-processing.

### 3.3.3 NinaPro: Classifying Electromyography Signals

Our final classification task, **NinaPro**, moves away from the image domain to classify hand gestures indicated by electromyography signals. For this, we use a subset of the NinaPro DB5 dataset [2] in which two thalmic Myo armbands collect EMG signals from 10 test individuals who hold 18 different hand gestures to be classified. These armbands leverage data from muscle movement, which is collected using electrodes in the form of wave signals. Each wave signal is then sampled using a wavelength and frequency prescribed in [5] to produce 2d signals.

**Data pre-processing:** Containing less than 4,000 samples, the data is comprised of single-channel signals with an irregular shape of 16*52 pixels. This task also differs from CIFAR for its class imbalance, as over $65\%$ of all gestures are the neutral position. We split the data using the same ratio as CIFAR, resulting in 2638 samples for training, and 659 samples for validation and testing each. No additional pre-processing is performed.

### 3.3.4 Darcy Flow: Solving Partial Differential Equations (PDEs)

Our first regression task, **Darcy Flow**, focuses on learning a map from the initial conditions of a PDE to the solution at a later timestep. This application aims to replaced traditional solvers with learned neural networks, which can output a result in a single forward pass. The input is a 2d grid specifying the initial conditions of a fluid and the output is a 2d grid specifying the fluid state at a later time, with the ground truth being the result computed by a traditional solver.

**Data pre-processing:**   we use scripts provided by [19] to generate the PDEs and their solutions, for a total of 900 data points for training, 100 for validation, and 100 for testing. All input data is normalized with constants calculated on the training set before fed into the neural network and de-normalized following an encode-decode scheme. The solutions, or labels, for the training set are also encoded and decoded this way. The test labels are not processed. We report the mean square error (MSE or $\ell_2$).

### 3.3.5 PSICOV: Protein Distance Prediction

Our final task, **PSICOV**, studies the use of neural networks in the protein folding prediction pipeline, which has recently received significant attention to the success of methods like AlphaFold [12]. While the dataset and method they use are too large-scale for our purposes, we consider a smaller set of protein structures to tackle the specific problem of inter-residual distance predictions outlined in [1]. 2d large-scale features are extracted from protein sequences, resulting in input feature maps with a massive number of channels. Correspondingly, the labels are pairwise-distance matrices with the same spatial dimension.

**Data pre-processing:**   we adopt the chosen subset of DeepCov proteins in [1], consisting of 3,456 proteins each with 128*128 feature maps across 57 channels. 100 proteins from this set are used for validation and the rest for training. Test data for final evaluation is gathered from another set of 150 proteins, PSICOV. Since these produce feature maps that are larger (512*512), we run the prediction network over all of its non-overlapping 128*128 patches. The evaluation metric is mean absolute error (MAE or $\ell_1$) computed on distances below 8 Å, referred to as $\text{MAE}_8$.

### 3.4 Ethics and Responsible Use

Within our array of tasks, the only dataset containing human-derived data is NinaPro. Our chosen subset of NinaPro contains only muscle movement data from 10 healthy individuals, without any exposure of personal information from clinical data. The original experiments to acquire NinaPro data are approved by the ethics commission of the state of Valais, Switzerland [2]. For other datasets, we have listed the data licenses in the appendix for responsible usages of data. While we do not view the specific datasets we use in this benchmark as potential candidates for misuse, the broader goal of applying NAS to new domains comes with inherent risks that may require mitigation on an application-by-application basis.

## 4 Using NAS-Bench-360 to Study Architecture Search Methods

Having detailed our construction of NAS-Bench-360, we now demonstrate its usefulness on (a) comparing and evaluating state-of-the-art architecture search methods on powerful search spaces and (b) discovering new insights on their performance on under-explored domains. In this section, we first specify the different NAS algorithms and baselines we compare, followed by the experimental and reproducibility setup we follow. Finally, we report our main comparisons and analyze the results.

### 4.1 Baselines and Search Procedures

From the discussion in Section 3, the two non-NAS baseline methods we consider—applying a tuned WRN to all tasks and using a fixed, domain-specific architecture—can be viewed via the NAS setup as having a singleton architecture search space. As for NAS algorithms themselves, we focus on two well-known paradigms for search: cell-based NAS (using DARTS [20]) and macro NAS (using DenseNAS [9]). We detail these four approaches below.

**Wide ResNet with Hyperparameter Tuning**    The residual network (ResNet) and its derivative architectures are canonical for classic computer vision, and we investigate their ability to generalize to our selection of tasks. A more powerful adaptation of ResNet, the Wide ResNet [31] is chosen as the backbone architecture. For automated training, we wrap the training procedure with a hyperparameter tuning algorithm, ASHA [15], an asynchonous version of Hyperband [18]. Given a range for each hyperparameter, either discrete or continuous, ASHA uniformly samples configurations and uses brackets of elimination: at each round, each configuration is trained for some epochs, before the algorithm selects the best-performing portion based on validation metrics. Since we use the Wide ResNet backbone for all tasks, our tuning budget is fixed and uniform.

**Expert-Designed Networks**    We also include expert-driven design of architectures in specific domains as a more rigorous comparator for NAS methods on our tasks. Frequently this includes not only hand-designed topologies and operation patterns but custom neural operations themselves, which are often crucial for success on domains beyond computer vision. Below we briefly summarize the architectures chosen for each task.

- **CIFAR-100**: While this task is very heavily studied and one can achieve very high accuracies using optimization tricks and transfer from ImageNet, we restrict our selection to existing results that use only the simple (standard) data augmentation we allow for the evaluation phase. Here the best result found is using DenseNet-BC [10].

- **Spherical**: This task is often regarded as a canonical example where a specific neural operation, specifically spherical convolutions, are the "right" operation to substitute for the convolution due to data-specific properties. Our result is from a wide variation of the spherical CNN in [4], with a max width of 256 channels from 64.

- **NinaPro**: As the original paper studying NinaPro used fairly weak networks that achieve a much higher error, here we simply report the performance of our tuned WRN baseline.

- **Darcy Flow**: Here we report the performance of a four-layer network that replaces convolutions with Fourier Neural Operators (FNOs) [19], which were specially designed solving partial differential equations. Note that our reproduced result attains slightly better MSE than the numbers reported by the authors.

- **PSICOV**: We report the reproduced performance of the ResNet-256 network used by the PDNET, a deeper, narrow, and dilated version of the standard ResNet used for ImageNet; note our reproduction attains much better $MAE_8$ than the authors report [1].

**Cell-based Search Using DARTS**    The first state-of-the-art NAS paradigm within our consideration is cell-based NAS. Cell-based methods first search for a genotype, which is a cell containing neural operations such as convolution and pooling. During evaluation, a neural network is constructed by replicating the searched cell and stacking them together. The most popular search space for this approach is the one used by the DARTS space [20], consisting of assigning one of eight operations to six edges in two types of cells: "normal" cells preserve the shape of the input representation while "reduction" cells downsample it. Note that for the dense tasks we do not use the reduction cell so as to not introduce a bottleneck.

Finally, to adhere to standard ML practices we do *not* adapt the standard DARTS pipeline, which uses test performance to select from multiple random seeds. This, in addition to not using other evaluation-time enhancements such—specifically auxiliary towers and the cutout data augmentation— leads to lower performance on CIFAR-100 than is reported in the literature. As this search space has been heavily studied since its introduction, we use as a search routine a recent approach—GAEA PC-DARTS—that achieves some of the best-known results on CIFAR-10 and ImageNet for this benchmark [16].

**Macro NAS Using DenseNAS**    The second NAS paradigm we consider is macro NAS. Instead of building from a fixed cell, macro NAS requires the specification of a super network with different inter-connected network blocks. These blocks and connections are then pruned during the search phase to construct the output neural net for evaluation. For this benchmark, we also choose a recent search space in this NAS paradigm, DenseNAS [9], which similarly to the DARTS space has near state-of-the-art results on ImageNet.

Table 2: Comparing NAS methods with baseline and expert-designed methods on NAS-Bench-360. All automated results (WRN, DenseNAS, and GAEA PC-DARTS) are averages of three random seeds. See Appendix for standard deviations.

| Search space | Search method | CIFAR-100 (0-1 err.) | Spherical (0-1 err.) | NinaPro (0-1 err.) | Darcy Flow MSE | PSICOV $MAE_8$ |
|---|---|---|---|---|---|---|
| WRN baseline | ASHA | 24.89 | 88.45 | **6.88** | 0.041 | 5.71 |
| expert design* | hand-tuning | **17.17** | 64.42 | **6.88** | **0.0096** | 3.50 |
| DenseNAS-R1 | DenseNAS | 27.44 | 72.99 | 10.17 | 0.10 | 3.84 |
| DARTS Cell | GAEA PC-DARTS | 24.19 | **52.90** | 11.43 | 0.056 | **2.80** |

\* Chosen according to best-effort literature search and implementation; c.f. Section 4.1.

DenseNAS searches for architectures with densely-connected, customizable routing blocks to emulate DenseNet [10]. In our experiments, we use the ResNet-based search space, DenseNAS-R1, with all of WRN's neural operations for better comparison with the baseline backbone. For point tasks and dense tasks, we adapt two super networks from the one used for ImageNet as inputs to the search algorithm. The super network for dense tasks maintains the same spatial dimensions without downsampling to avoid bottlenecks, and we use a lower learning rate for evaluating architectures on dense tasks to prevent divergence. Other training and evaluation procedures are identical to those in the original paper and uniform across all tasks.

## 4.2 Experimental Setup

Our main experiments consist of 3 evaluation trials for every combination of method and task, fixing one random seed for each trial. We present these results in Table 2 and discuss the specific procedure, reproducibility, and extension experiments in the following subsections.

**Using validation data**   For best practices in NAS, we argue for the separation of the final testing set and the validation set, which is specifically for selecting neural architectures and hyperparameters. After this process, we combine training and validation data to perform retraining and evaluation on the test set. This result is reported as final and is not used in any way to further optimize the model.

**Hyperparameter tuning**   In experiments with hyperparameter tuning, we consistently use the same hyperparameter ranges and fix the tuning budget, in terms of the number of configurations and maximum training epochs, across all tasks. The tuning budget is selected to be 2.5 to 3 times the backbone training time. This is to eliminate inductive biases for specific tasks. Details on the tuning procedure are in the appendix.

**Software and hardware**   We adopt the free, open-source software *Determined*[1] for experiment management, hyperparameter tuning, AWS cloud deployment with docker containers. All experiments are performed on a single p3.2xlarge instance with one Nvidia V100 GPU. The computation cost in GPU hours of individual experiments using this setup can be found in the appendix.

**Reproducibility**   The following measures in our experimental pipeline are taken to ensure the reproducibility of our results:

1. We perform most data pre-processing steps beforehand and store the processed data in the cloud for download. A data splitting scheme, once randomly selected, is then fixed for all experiments on that task, i.e. the same training, validation, and testing sets fed into the dataloader are always the same.

2. Experimentation code is always executed in a fixed docker container using a pre-built docker image on Docker Hub. This guarantees a uniform execution environment and saves users from the manual labor of configuring dependencies.

---

[1]GitHub repository: `https://github.com/determined-ai/determined`

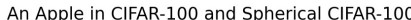
An Apple in CIFAR-100 and Spherical CIFAR-100

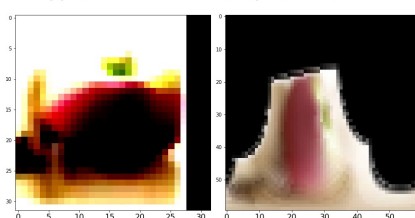

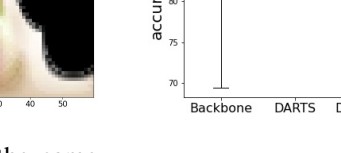

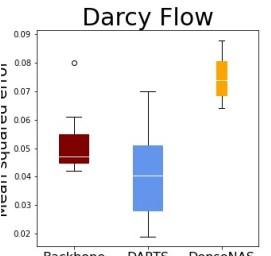

Figure 1: Comparison of the same CIFAR-100 image before and after the spherical transformation.

Figure 2: Distribution of random architectures and hyperparameters' performance on NinaPro and Darcy Flow.

Table 3: Experiment runtimes of NAS-Bench-360 (GPU hours)

| Task | GAEA PC-DARTS | DenseNAS | WRN |
|---|---|---|---|
| CIFAR-100 | 33 | 2 | 8 |
| Spherical CIFAR-100 | 39 | 2.5 | 8.5 |
| NinaPro | 2 | 0.5 | 1 |
| Darcy Flow | 15 | 0.5 | 2 |
| PSICOV | 59.5 | 23 | 61 |

3. Via the specification of a random seed, *Determined* controls several important sources of randomness during code execution, including hyperparameter sampling and training data shuffling.

4. During training, we always validate on the full validation set, not on a mini-batch, to avoid stochasticity in the results.

### 4.3 Comparing NAS Approaches Using NAS-Bench-360

**Generalization of NAS to other domains**    Our experiments demonstrate that state-of-the-art NAS approaches in classic vision are unable to outperform human-designed neural networks on 3 out of 5 tasks in NAS-Bench-360. They do especially poorly on the Darcy Flow task and fall short of matching both non-NAS comparators by a large margin. Perhaps most surprisingly, neither the DARTS space nor DenseNAS, both very recent search spaces with strong results on ImageNet (and CIFAR-10 for the former) are able to outperform the reported performance of a fairly basic architecture (DenseNet) on CIFAR-100; this is especially interesting as DenseNAS was built around this architecture. Overall, our results suggest that modern NAS, despite its promise to automate deep learning, is not yet well-equipped to handle its various domains of applications studied in this paper. These empirical results also serve as new baselines for comparison in future research to extend NAS approaches to generalize to new areas.

**Computational cost**    In some time-sensitive applications of NAS, both efficiency and performance are criteria for NAS method selection. Our choice of methods exemplifies a tradeoff between these two factors. As a more computationally heavy method, GAEA PC-DARTS beats the more lightweight DenseNAS on most of the tasks except for NinaPro, where they achieve similar accuracies. On certain tasks, such as NinaPro and PSICOV, DenseNAS would be the more cost-effective option than GAEA PC-DARTS to have decent performance on par with handcrafted neural architectures. Note that the computation cost of the WRN baseline can vary due to randomness inherent in ASHA's asynchrony. We report all experiment runtimes in Table 3.

**CIFAR-100 vs. Spherical**    The Spherical task can be directly compared to CIFAR-100 to assess how well NAS methods could handle image distortions. With the same setup across tasks, both

Table 4: $\ell_1$ error of supernet and searched architectures (discretized) on grid tasks

| | DARTS | | DenseNAS | |
| Task | Supernet | Discretized | Supernet | Discretized |
| --- | --- | --- | --- | --- |
| Darcy Flow | $0.031 \pm 0.001$ | $0.057 \pm 0.012$ | $0.041 \pm 0.002$ | $0.10 \pm 0.010$ |
| PSICOV | $3.87 \pm 0.12$ | $2.80 \pm 0.057$ | $7.96 \pm 0.20$ | $3.84 \pm 0.15$ |

the DARTS space and DenseNAS have reasonably good numbers on CIFAR-100, but their results significantly deteriorate on the spherical variant. Both obtain much worse error when the images are spherically projected, but a much larger gap emerges between the two methods, with DenseNAS performing quite badly. On the other hand, the searched DARTS Cell not only performs 20-36% better than the other convolutional approaches but even beats our best-effort adaptation of the spherical CNN approach to this task [4], in which we expanded the size of that network. This is surprising because spherical convolutions were designed specifically for such data. We believe these results indicate that the spherical dataset may be a useful but simple way for distinguishing NAS approaches when they are overfitting to standard computer vision domains; Figure 1 provides an example of the distortion.

**WRN as a baseline**   Viewing the WRN baseline as a singleton architecture search space, we compare this baseline to more sophisticated NAS search spaces. On our set of new tasks, NAS does not perform better than Wide ResNet with hyperparameter tuning on CIFAR-100, NinaPro, and Darcy Flow but excels on the rest. Hyperparameter optimization can boost the backbone performance considerably to rival the performance of NAS methods. Most non-Bayesian hyperparameter tuning algorithms, such as random search [17], population-based training [11], and Hyperband [18], are also straightforward to apply with any neural network backbone. Therefore, we argue for the use of hyperparameter-tuned backbones to assess the effectiveness of NAS approaches and encourage their inclusion in NAS benchmarks.

## 4.4   In-Depth Studies Using NAS-Bench-360

**Supernet performance on grid tasks**   During architecture search, our NAS methods on the DARTS and DenseNAS search spaces train the supernets to find optimal neural operations on the validation set. Surprisingly, the validation error of the supernet is sometimes lower than that of the final searched, discretized neural network. Therefore, we evaluate the supernet of DARTS and DenseNAS on the testing set, and we compare its performance with that of the final neural network in Table 4. The supernet outperforms the final network on Darcy Flow for both methods, but the reverse is true for the PSICOV task and all point tasks. The supernet is not in the search space and so we report the discretized result; nevertheless, this fact suggests that performance on a task like Darcy Flow might benefit from a better search space.

**Evaluating random architectures and hyperparameters**   The power of an architecture or hyperparameter space can also be characterized by the performance of its random elements. We assess both the average and variance of the results. To do this, we randomly sample 8 network architectures each from the search spaces of DARTS and DenseNAS, and we test their performance on the NinaPro and Darcy Flow tasks, one for classification and the other for regression. For comparison, we also randomly sample 8 hyperparameter configurations to train the backbone Wide-ResNet in Figure 2. While rather successful on NinaPro, the random architectures have a high average error and vary in performance on the Darcy Flow task. Random hyperparameters are more unstable on NinaPro, but its median performance is better than NAS.

**Utility of hyperparameter tuning**   The final experiment examines whether hyperparameter tuning improves the performance of WRN on various tasks. During hyperparameter search, we compare the validation metrics of training using default hyperparameters and using tuned ones from ASHA to select final hyperparameters for retraining. Despite the small tuning budget allocated to ASHA, tuned hyperparameters could outperform the default setting on all tasks except for CIFAR-100. Our results suggest that wide ResNet's standard set of hyperparameters are only optimized for conventional image classification. On other tasks, hyperparameter optimization is helpful for boosting performance.

## 5   Conclusion

**NAS-Bench-360** is a benchmarking suite with a novel, diverse set of tasks. The tasks are derived from various fields of academic research, leading to different potential applications. Our selection of NAS approaches achieves state-of-the-art performances on most tasks, which points to new possibilities of incorporating NAS into new research domains. All datasets and reproducible experiment code are open-sourced, and we welcome researchers to use these tasks and further iterate on them with new NAS methods. Finally, a possible extension to generalize this set of tasks is datasets with 1d or 3d inputs, such as audio. We hope our work can encourage the NAS community to move towards tackling more diverse problems in the real world.

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
