# OpenReview forum: "NAS-Bench-360: Benchmarking Diverse Tasks for Neural Architecture Search"
_NeurIPS.cc/2021/Track/Datasets_and_Benchmarks/Round1 — Submitted to NeurIPS 2021 Datasets and Benchmarks Track (Round 1)_

### Official Review · Reviewer_wS9G · 2021-06-26
**Well-written paper with interesting results; the datasets are not very diverse**

**Rating:** 5
**Confidence:** 4
**Correctness:** No concern.
**Clarity:** The paper is well-written and clear.

**Strengths:**

1.	Introducing diverse tasks to better evaluate the performance is valuable as it can help facilitate the research of NAS to tackle different tasks.
2.	The experimental parts are well-written, with sufficient details for reproducing the results, and various observations.
3.	The code and datasets are published on an official website and Github, and the authors promised to maintain the project.

**Weaknesses:**

1.	I am disappointed that, while the main motivation of the paper is diversity, the benchmark only has 5 datasets (with 4 new datasets) and seem to have low coverage of the tasks. There are a bunch of other tasks that could be considered, such as graph data (node classification, graph classification, etc.), time-series (prediction, classification, etc.), texts (various NLP tasks), video tasks, tabular tasks, and other applications (e.g., recommender systems).
2.	It is well-known that different application domains require different search spaces. The authors just simply reuse the search space that is designed for image classification. Thus, it is not surprising to observe the expert-designed architecture is better.
3.	As of the date I review the paper, the README of GitHub seems very short (less than 10 lines of descriptions) without enough instructions. It is unclear to me how to install the benchmark, how to reproduce the results. I am not clear whether and how I can run the benchmark on my own server either, as it seems that we can only run the code on AWS.


**Additional Feedback:**

Some additional questions:

1.	“PDE” is never explained. What does “PDE” stand for?
2.	Why is the benchmark called “360”?


**Documentation:**

The authors provided sufficient details for the datasets. However, more descriptions could be added to Github and the official website.

**Ethics:**

No concern.

**Relation To Prior Work:**

The authors provided sufficient related work

**Summary And Contributions:**

This paper proposes Nas-Bench-360, a benchmark suite for evaluating state-of-the-art NAS methods. The paper argues that we need to evaluate NAS methods on a suite of diverse benchmarks. Following this, the authors introduce 4 tasks, covering different applications. Further, the authors benchmark different search spaces on the datasets. It observed that many NAS methods may fall short in new applications compared with the expert-designed architectures.

---

### Official Review · Reviewer_n5AH · 2021-07-01

**Rating:** 4
**Confidence:** 4
**Correctness:** See above

**Strengths:**

The idea of evaluating NAS on a different set of problem domains is interesting.
Unfortunately, I think the execution could have been improved.

**Weaknesses:**

When I want to deploy a NAS algorithm in practice, I want to obtain a high quality model with little human effort. To me this implies that a high quality NAS method should be able to do the following:
1. Does the NAS method find a good architecture in the search space
2. Does the NAS approach find a good architecture more efficiently than Random search
3. Does the NAS approach find an architecture that is better (complexity/quality tradeoff) than simply picking the largest model in the search space.

Finally if it is successful in 1,2,3 success still depends on the following:
4. Is the search space suited to the problem at hand.


Regarding 1 and 2
The paper does not provide a random baseline for all settings (it does for 2). It also does not provide a high quality model in the actual search space. Therefore it is not clear to me whether the NAS method found a poor result or whether the search space was not a good fit for the problem.

Regarding 3:
It is unclear to me what the cost is (#FLOPS/#Parameters/#Actual latency/#Training time/....) of the expert models and WRN vs the NAS models. It might be that the model capacity explains the difference in quality.

Regarding 4:
It is not clear to me whether the failure of the NAS methods in certain cases is due to the search space design or due to the algorithm.

Since

**Additional Feedback:**

Everything should be contained in the boxes above.

**Clarity:**

- The renaming of classification tasks and regression tasks to point and grid tasks does not make sense.

**Documentation:**

No new datasets

**Ethics:**

No ethical concerns

**Relation To Prior Work:**

This seems fine.

**Summary And Contributions:**

The paper is motivated by the fact that NAS methods are typically evaluated on ImageNet/Cifar 10 but it is not clear how well they work on other types of data. For that reason the authors propose to evaluate the NAS methods on Cifar-100, A modified variation of cifar-100, EMG data, a PDE problem and a protein folding problem.
The datasets used in this benchmark are already publicly available but they are now used for NAS evaluation.

Therefore the main contribution is the analysis. Unfortunately the contribution leaves me with multiple questions and I think that this could be improved.

---

### Official Review · Reviewer_qWDt · 2021-07-05
**Interesting study but short of in-depth analysis**

**Rating:** 3
**Confidence:** 5
**Correctness:** Correct.
**Clarity:** Well written.

**Strengths:**

1. The studied problem is well motivated.
2. Adequate experiments are conducted to study the problem.
3. The paper is well written.


**Weaknesses:**

1. The benchmark is biased in term of scale. Five datasets are included in the benchmark, though covering a fair breadth of domains, the datasets are all pretty small, limiting the generalizability of conclusions drawn based on the experiments.

2. The study is kind of trivial. The datasets are easy to prepare. Some simple baseline methods were run on the datasets and the experimental results were reported. The analysis are far from in-depth (almost all results are shown in Table 2). The conclusions are also not attractive. I do not think it is good enough to have a paper published here by running some simple methods on several dataset of limited scale and reporting the experimental results.

3. It is strange to put the main manuscript as the major part of the supplementary material.

**Additional Feedback:**

After reading the rebuttal, I am still concerned about the diversity of this benchmark. So, I would hold my score.

**Documentation:**

Yes, it is.

**Ethics:**

No, good enough.

**Relation To Prior Work:**

It  is clear.

**Summary And Contributions:**

This paper aims to expand to benchmarking effort of NAS to tasks more than the widely-used vision tasks. A new benchmark which assembles five datasets of different domains is prepared. Four methods, including both hand-crafted models and NAS models,  are executed on the benchmark. Some conclusions are drawn from the experimental results.

---

### Decision · Program_Chairs · 2021-07-26

**Decision:**

Reject

**Comment:**

Studying NAS for more diverse tasks and search spaces is a very important direction of the field.
However, for the current state of the work, all reviewers gave rejection scores for diverse reasons and actively kept them after the rebuttal. Taken together, these reasons also make me recommend rejection.